# Coronary Artery Bypass in Young Patients—On or Off-Pump?

**DOI:** 10.3390/jcm8020128

**Published:** 2019-01-22

**Authors:** Ryoi Okano, Yi-Jia Liou, Hsi-Yu Yu, I-Hui Wu, Nai-Kuan Chou, Yih-Sharng Chen, Nai-Hsin Chi

**Affiliations:** 1Department of Surgery, National Taiwan University Hospital and National Taiwan University College of Medicine, Taipei 10002, Taiwan; ryoi.okano@gmail.com (R.O.); hsiyuyu@gmail.com (H.-Y.Y.); aaronihuiwu@gmail.com (I-H.W.); nickchou@ntu.edu.tw (N.-K.C.); yschen1234@gmail.com (Y.-S.C.); 2Department of Cardiovascular Surgery, Ageo Central General Hospital, Saitama 362-8588, Japan; 3Department of Life Science, National Dong Hwa University, Hualien 97401, Taiwan; conlyc7779@gmail.com

**Keywords:** coronary artery bypass, on-pump coronary artery bypass, off-pump coronary artery bypass, young patients

## Abstract

A definitive conclusion regarding whether on-pump or off-pump coronary artery bypass is preferable in young patients is lacking. The aim of our study was to perform a long-term comparison of the two approaches in young patients. We analyzed the National Health Insurance Research Database, using data for patients between 18 and 45 years of age who had undergone isolated coronary artery bypass between 2001 and 2011. The study endpoints were: all-cause death, major adverse cardiac and cerebrovascular events, and repeat revascularization within 30 days, 1 year, 5 years, and the entire 10-year follow-up period. A total of 344 patients received off-pump surgery and 741 patients received on-pump surgery. Preoperative characteristics and comorbidities were similar in both groups, and all-cause mortality was almost equal (*p* = 0.716). The 5-year survival rates were 93.9% and 92.2% in the off-pump and on-pump groups, respectively, and the 10-year survival rates were 86.3% and 82.1%, respectively. The repeat revascularization rate was significantly lower in the on-pump group (*p* = 0.0407). Both the on-pump and off-pump methods offer equally good long-term outcomes in terms of mortality and major adverse cardiac and cerebrovascular events. However, the need for repeat revascularization is a concern in the long term after off-pump surgery.

## 1. Introduction

The efficacy and appropriate indications for off-pump coronary artery bypass grafting (CABG) have been a source of contention since its enthusiastic re-emergence in the 1990s [1]. Recently, several large randomized trials have compared on-pump with off-pump CABG [2,3,4,5,6]. They showed the superiority, or at least non-inferiority, of on-pump CABG to off-pump CABG for most patients and in the hands of most surgeons [7]. However, the main participants in these trials were elderly patients. In most of the trials the average age was approximately 60 years, and in some trials patients under 45 years were even excluded [2,3,4,5,6].

Young patients account for 3% of all patients with coronary artery disease (CAD) and have a different background and propensity to elderly patients [8]. They are more likely to be male, smokers, obese, have a family history, and some of them have non-atherosclerotic diseases [8,9,10,11,12,13,14,15,16,17]. Although the appropriateness of CABG in young patients is currently being considered [8,9,10,11,12,13,14,16], there is insufficient data to form a definitive conclusion. In particular, there have been no studies comparing long-term outcomes between on-pump and off-pump CABG in young patients.

The aim of our study was to perform the first long-term, high-volume comparison of on-pump versus off-pump CABG in young patients using the Taiwan Nationwide Database.

## 2. Experimental Section

### 2.1. Materials and Methods

The Taiwanese National Health Insurance (NHI) program has been operating since 1995. By 2014, approximately 99.9% of the Taiwanese population was enrolled in the NHI program [18]. For the current analysis, we used a data subset of the NHI Research Database, namely the registry for patients with catastrophic illness, which contains the records of all prevalent cases of patients who have major illnesses. 

From the database, we included patients between 18 and 45 years of age who had undergone isolated on-pump or off-pump CABG between 2001 and 2011, as shown in Figure 1. We searched the database to determine the presence of comorbidities such as hypertension, diabetes mellitus, peripheral vascular disease, chronic obstructive pulmonary disease stroke, myocardial infarction (MI), implanted cardiac pacemaker, or atrial fibrillation. The extent of CAD (single, double, or triple vessel disease) and renal function were also compared. The study endpoints were all-cause death, major adverse cardiac and cerebrovascular events (MACCE; defined as all-cause death, MI, and stroke); and repeat revascularization within 30 days, 1 year, 5 years, and over the entire follow-up period. 

### 2.2. Statistical Analysis

The characteristics of patients were analyzed using the chi-square test for categorical variables and *t*-tests for continuous variables to identify potential differences between the two groups. We used standardized difference to measure the balance of covariates, whereby an absolute standardized difference of greater than 10% represented meaningful imbalance.

The survival was compared between the groups of patients using Kaplan–Meier survival curves and log-rank tests. The factors affecting survival were analyzed by univariate and multivariate Cox’s proportional hazard models to estimate the hazard ratio (HR) with a 95% confidence interval (95% CI). All the analyses were performed with SAS 9.3 software (SAS Institute Inc, Cary, NC, USA). A *p*-value <0.05 was considered statistically significant.

## 3. Results

There were 31,448 patients that had undergone CABG in the database. From these, we included 1282 patients who were 45 years old or younger and had isolated on-pump or off-pump CABG. Among these, 197 patients were excluded due the use of an on-pump beating heart technique, as shown in Figure 1. The average overall follow-up time was 5.3 years and the longest was 16 years, with 100% complete follow up. The mean follow-up times were 4.79 years for off-pump CABG and 5.53 years for on-pump CABG. 

A total of 344 patients received off-pump surgery and 741 patients received on-pump surgery, as shown in Table 1. Both groups were comprised of predominantly male patients, and mean age was similar in both groups (approximately 41 years in both; *p* = 0.18). The distribution of one-vessel disease was greater in the off-pump group (15.99% off-pump vs. 6.34% on-pump). Before the operation, both groups had a similar incidence of diabetes mellitus, hyperlipidemia, familial hypercholesterolemia, chronic obstructive pulmonary disease, stroke, peripheral vascular disease, hypertension, previous myocardial infarction, atrial fibrillation, implanted pacemaker, and presence of renal dysfunction, as shown in Table 1. In the off-pump group, there were 28.19% of them that received previous percutaneous coronary intervention and in the on-pump group it was 26.85% (*p* = 0.63). The emergent operations of both groups were similar; they accounted for 13.95% in the off-pump and 13.76% in the on-pump group (*p* = 0.41). 

Regarding the completeness of revascularization, we used graft number/disease vessels as a marker; if the number is more than 1, we deem this as complete revascularization in our database. By doing so, the completeness of revascularization in one vessel disease was 100% in both groups. The completeness of revascularization in two-vessel disease was 93% in the off-pump group and 95% in the on-pump group. The completeness of revascularization in three-vessel disease was 91% in the off-pump group and 92% in the on-pump group.

The 30-day outcomes, as shown in Table 2, and the 1-year outcomes, as shown in Table 3, in terms of death, myocardial infarction, and stroke were not different between the groups. The operative outcome, as shown in Table 2, including transfusion amount, post-operative atrial fibrillation, ICU (intensive care unit) stay, and hospital stay were no different. The incidence of post-operative atrial fibrillation was 10.46% in the off-pump and 13.76% in the on-pump groups (*p* = 0.535). Further, at 5 years and 10 years, there were no differences in death, MI, stroke, and MACCE, as shown in Table 4 and Table 5. The Kaplan–Meier curve shows that the rates of all-cause mortality were almost equal in both groups (*p* = 0.716), as shown in Figure 2A. The 5-year survival rate was 93.9% in the off-pump group and 92.2% in the on-pump group and the 10-year survival rates were 86.3% and 82.1%, respectively. There was no significant difference in freedom from MACCE at 5 and 10 years in patients receiving on-pump versus off-pump surgery (79.1% vs. 81.8% and 64.8% vs. 70.0%, respectively; *p* = 0.3310), as shown in Figure 2B.

Repeat revascularization was defined by further admission intervention of coronary artery or repeat coronary artery bypass surgery 6 months after first operation. The reason we defined repeat revisualization by 6 months after first operation was to eliminate the scheduled hybrid procedures, because most of the hybrid procedures were performed within 3 months. 

The repeat revascularization rate was significantly lower in the on-pump group (*p* = 0.0407), as shown in Figure 2C. Freedom from revascularization at 5 years was 86.2% in the on-pump versus 80.8% in the off-pump group, and at 10 years was 76.1% versus 65.9%, respectively.

## 4. Discussion

In our study, we compared the outcomes of on-pump and off-pump CABG in 1085 young patients. There was no difference in the 30-day, 1-year, and 5-year outcomes in terms of death, MI, stroke, and new renal failure. Kaplan–Meier and long-rank analysis showed that all-cause mortality and MACCE were not significantly different. Off-pump surgery was not inferior to on-pump surgery in terms of death, MI, and MACCE. However, the repeat revascularization rate was significantly higher in the off-pump group. The long-term survival was excellent in both groups. 

For young coronary artery bypass patients, the long-term durability of the vascular graft and the long-term survival benefit are the major concerns. Young patients have less comorbidities than older patients. They have better renal function, less incidence of diabetes, and less renal impairment compared with older patients. Therefore, in both groups, young patients have less surgical risk [19].

The characteristics of coronary artery disease in young patients are different from older patients: 80% is atherosclerotic and 20% is non-atherosclerotic (e.g., coronary embolism, thrombosis, congenital anomalies, vessel inflammation, spasm) [9,10,11,12,13,14,15,16,17].

Compared with older patients, coronary heart disease in young patients is more closely associated with male sex, smoking, obesity, and family history. Conversely, the proportion of patients with diabetes mellitus and hypertension is smaller [8,9,10,11,12,13,17]. In our patients, the prevalence of diabetes was 33.72% and 31.85% in the off-pump and on-pump groups, respectively; in contrast, in our previous nationwide study where the mean age of patients was 62 years, the incidence was approximately 70% [19]. The overall prevalence of hypertension was 55%, and 90% of the cohort were males.

Young patients with coronary artery disease have fewer comorbidities than older patients [10,12]. In our study, the incidence of stroke, peripheral vascular disease, chronic obstructive pulmonary disease, and atrial fibrillation were low, and renal function was preserved in most patients.

Triple vessel disease was the predominant pathology in our cohort, in accordance with similar findings reported recently by Saraiva et al. [10]. 

Kaplan–Meier analysis in this study revealed excellent survival in young patients, and there were no significant differences between on-pump and off-pump CABG. However, the repeat revascularization rate was significantly higher after off-pump CABG. As recognized in the European Society of Cardiology/European Association for Cardiothoracic Surgery 2014 guidelines, there is substantial evidence indicating that on-pump CABG provides superior or equal short-and-long term outcomes compared with off-pump CABG in elderly patients [2,3,4,5,6,7]. However, the more frequent need for revascularization may be a major concern when treating younger patients who would be expected to live longer, and their rates of revascularization may keep rising in parallel to their follow-up.

Previous studies had suggested that the higher rate of repeat revascularization in off-pump CABG patients was related to incomplete revascularization and the surgeon’s experience [20]. In addition, the rapid progression of atherosclerosis in young patients [15] might also have precipitated the high rate of repeat revascularization in our study. Regarding the completeness of revascularization, we used graft number/disease vessels as a marker; if the number is more than 1, we deem this is complete revascularization in our database. By doing so, the completeness of revascularization in one vessel disease was 100% in both groups. The completeness of revascularization in two-vessel disease was 93% in the off-pump group and 95% in the on-pump group. The completeness of revascularization in three-vessel disease was 91% in the off-pump group and 92% in the on-pump group. In our national database, the completeness of revascularization in coronary artery patients was more than 91% in all, and there were no differences in both groups. However, we cannot trace specific surgeon’s experience in the registry database. Presuming that both groups have an equal completeness revascularization rate, we could say that the repeat revascularization was not due to incomplete surgery at the first operation. The differences might come from the procedure itself.

Actually, both methods offer good immediate results and provide also good long-term outcomes in terms of death, myocardial infarction, and stroke. We thought the choice of off-pump and on-pump procedures in the young patient should be decided by the patient’s lesion anatomy, and the surgeon’s experience and preference. 

Several possible limitations to our study must be addressed. Because the study analysis was retrospective and only clinical events and services regulated by reimbursement were recorded, factors such as the extension of diseased coronary arteries and the completeness of revascularization, left ventricular ejection fraction or patients’ functional capacity, choice of conduit (percentage of bilateral internal mammary arteries or total arterial grafting), conversion between on-pump and off-pump, and surgeon experience were not evaluated. The decision to perform off-pump versus on-pump CABG was dependent on the patients’ clinical characteristics and the surgeons’ preferences. Unmeasured confounders or detection bias may thus have affected our results.

## 5. Conclusions

We compared on-pump and off-pump CABG in young patients using a nationwide database. There was no significant difference in short-and-long term outcomes in terms of death and MACCE. However, the need for repeat revascularization might be a concern over the long term after off-pump surgery.

## Figures and Tables

**Figure 1 jcm-08-00128-f001:**
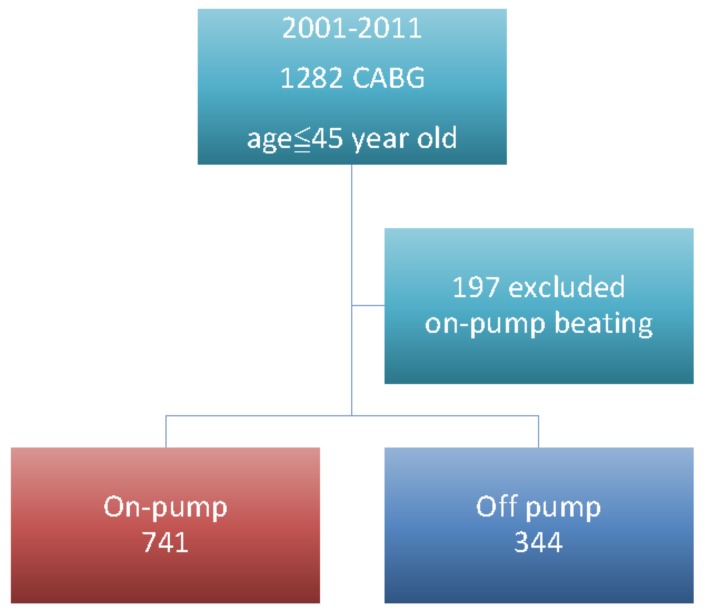
Flowchart of patient enrollment. We included 1282 patients, who were 45 years old or younger, and had isolated on-pump arrest or off-pump coronary artery bypass grafting (CABG). Among them, 197 patients were excluded for the surgeons’ adopted on-pump beating techniques. Mean follow up time was 5.53 years and longest follow up time was 16 years, with 100% complete follow up.

**Figure 2 jcm-08-00128-f002:**
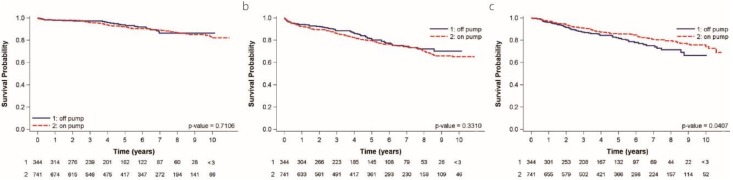
Kaplan–Meier survival curve. (**a**) All cause of death in both groups. The 5-year survival rate was 93.9% in the off-pump and 92.2% in on-pump groups; the 10-year survival rate was 86.3% in the off-pump and 82.1% in the on-pump groups. (**b**) Major cardiac and cerebrovascular events (death, myocardial infarction, and stoke). The 5-year free from major adverse cardiac and cerebrovascular events (MACCE) rate was 79.1% in the off-pump and 81.8% in the on-pump groups; the 10-year free from MACCE rate was 64.8% in the off-pump and 70.0% in the on-pump groups. (**c**) Kaplan–Meier survival curve of repeat revascularization. Off-pump carries concern of repeat revascularization in the long term. Free from revascularization at 5 years was 86.2% in on-pump vs. 80.8% in off-pump, and at 10 years, 76.1% vs. 65.9%, respectively.

**Table 1 jcm-08-00128-t001:** Baseline characteristics of the patients.

Characteristic	Age ≤ 45 Year (*N* = 1282)					
	Off-Pump CABG		On-Pump CABG		*P* Value	Standardized Difference (%)
	*N*	%	*N*	%		
Patients, total No.—no. (%)	344		741			
Female sex	34	9.88	94	12.69	0.18	8.86
Age—year Mean/Std. Dev.	41.04 ± 4.15		40.95 ± 4.13			2.17
Extent of coronary artery disease						
One-vessel disease	55	15.99	47	6.34	<0.0001	30.99
Two-vessel disease	72	20.93	113	15.25		14.80
Three-vessel disease	217	63.08	581	78.41		34.18
Number of grafting						
1	57	16.56	45	6.07	<0.0001	25.14
2	42	12.20	61	8.23		9.63
3 and >3	245	71.22	635	85.69		8.14
Comorbidity						
Diabetes Mellitus	116	33.72	236	31.85	0.54	3.99
Hyperlipidemia	65	18.89	131	17.67	0.62	4.21
Familial hypercholesterolemia	6	1.74	15	2.02	0.31	5.11
Insulin-dependent diabetes mellitus	8	2.33	7	0.94	0.07	10.90
Chronic obstructive pulmonary disease	8	2.33	16	2.16	0.86	1.12
Previous stroke	23	6.69	45	6.07	0.70	2.51
Peripheral vascular disease	9	2.62	19	2.56	0.96	0.33
Hypertension	195	56.69	400	53.98	0.40	5.44
History of myocardial infarction	107	31.10	214	28.88	0.45	4.86
History of atrial fibrillation	5	1.45	6	0.81	0.32	6.09
Implanted pacemaker	7	2.03	8	1.08	0.21	7.72
Normal renal function	322	93.60	698	94.20	0.70	2.48
Chronic kidney disease	5	1.45	8	1.08	0.60	3.34
Renal-replacement therapy	17	4.94	36	4.86	0.95	0.39
Prior percutaneous coronary intervention	97	28.19	199	26.85	0.63	5.23
Surgical Timing—emergency	48	13.95	102	13.76	0.91	0.41

CABG: coronary artery bypass grafting; Std. Dev.: Standard Deviation.

**Table 2 jcm-08-00128-t002:** Outcomes at 30 days.

Outcome	Off-Pump CABG	On-Pump CABG	Adjusted Gender and Age and Other Variables (with Stepwise Selection)			
			Hazard Ratio (On:off)			*P* Value
			HR	95% C.I.		
				Low	Up	
Operative outcome						
Transfusion—pRBC (unit)	2.56 ± 1.45	3.12 ± 1.63	0.93	0.34	2.42	0.672
Post-OP atrial fibrillation—no. (%)	36 (10.46)	102 (13.76)	1.32	0.41	2.16	0.535
ICU stay (days)	1.4 ± 0.82	1.5 ± 0.93	1.02	0.81	1.32	0.812
Hospital stay (days)	8.2 ± 3.58	9.1 ± 4.21	1.15	0.36	4.21	0.781
Components of primary outcome—no. (%)						
Death	2 (0.58%)	3 (0.4%)	1.43	0.15	13.75	0.7581
Myocardial infarction	4 (0.11%)	7 (0.94%)	0.84	0.25	2.87	0.7805
Stroke	2 (0.58%)	2 (0.28%)	0.41	0.03	6.63	0.5266
New renal failure requiring dialysis	5 (1.45%)	12 (1.62%)	1.10	0.39	3.14	0.8596
Other specified outcomes—no. (%)						
Cardiovascular-related death	1 (0.29%)	1 (0.13%)	0.46	0.03	7.29	0.5781

pRBC: Red blood cell; ICU: intensive care unit; CABG: coronary artery bypass grafting; HR: hazard ratio; C.I.: confidence interval.

**Table 3 jcm-08-00128-t003:** Outcomes at 1 year.

Outcome	Off-Pump CABG	On-Pump CABG	Adjusted Gender and Age and Other Variables (with Stepwise Selection)			
			Hazard Ratio (On:off)			*P* Value
			HR	95% C.I.		
				Low	Up	
Components of primary outcome—no. (%)						
Death	7 (2.03%)	12 (1.62%)	0.80	0.32	2.03	0.6392
Myocardial infarction	11 (3.2%)	36 (4.86%)	1.46	0.73	2.89	0.2819
Stroke	6 (1.74%)	11 (1.48%)	0.87	0.31	2.41	0.7879
New renal failure requiring dialysis	13 (3.78%)	35 (4.72%)	1.59	0.83	3.05	0.1622
Other specified outcomes—no. (%)						
Cardiovascular-related death	13 (3.78%)	11 (1.48%)	0.98	0.23	4.22	0.9738

**Table 4 jcm-08-00128-t004:** Outcomes at 5 years.

Outcome	Off-Pump CABG	On-Pump CABG	Adjusted Gender and Age and Other Variables (with Stepwise Selection)			
			Hazard Ratio (On:off)			*P* Value
			HR	95% C.I.		
				Low	Up	
Components of primary outcome—no. (%)						
Death	16 (4.65%)	45 (6.07%)	1.27	0.72	2.25	0.4109
Myocardial infarction	28 (8.14%)	67 (9.04%)	1.11	0.71	1.74	0.6420
Stroke	18 (5.23%)	37 (4.99%)	1.00	0.56	1.77	0.9909
New renal failure requiring dialysis	29 (8.43%)	68 (9.18%)	1.18	0.76	1.84	0.4650
Other specified outcomes—no. (%)						
Cardiovascular-related death	5 (1.45%)	13 (1.75%)	1.14	0.41	3.21	0.7996
Cerebrovascular disease death	2 (0.58%)	2 (0.27%)	1.18	0.10	13.64	0.8945

**Table 5 jcm-08-00128-t005:** Outcomes followed by death.

Outcome	Off-Pump CABG	On-Pump CABG	Adjusted Gender and Age and other Variables (with Stepwise Selection)			
			Hazard Ratio (On:off)			*P* Value
			HR	95% C.I.		
				Low	Up	
Components of primary outcome—no. (%)						
Death	25 (7.27%)	70 (9.45%)	1.15	0.73	1.83	0.5492
Myocardial infarction	35 (10.17%)	90 (12.15%)	1.12	0.75	1.67	0.5737
Stroke	22 (6.4%)	48 (6.48%)	0.99	0.60	1.65	0.9731
New renal failure requiring dialysis	32 (9.3%)	79 (10.66%)	1.14	0.75	1.72	0.5533
Other specified outcomes—no. (%)						
Cardiovascular-related death	5 (1.45%)	15 (2.02%)	1.30	0.47	3.58	0.6113
Cerebrovascular disease death	2 (0.58%)	2 (0.27%)	1.18	0.10	13.64	0.8945

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
