# Peer review of "Coronary Artery Bypass in Young Patients—On or Off-Pump?"

_jcm, 2019, doi:10.3390/jcm8020128_

Reviewer 1 Report

This retrospective study highlighted the long-term clinical results of the patients under 45 years of age undergoing isolated CABG, especially comparing the outcome between those with on-pump (probably with cardiac arrest) manner (n=741) and those with off-pump manner (n=344).

Preoperative comorbidities regarding dyslipidemia or familial hypercholesterolemia are missing along with too many factors including history of previous PCI, NHYA/CCS class, preoperative left ventricular function, and presence of heart valve lesions and some other factors as they have mentioned in the limitation section. These missing factors should have been also considered in multivariate analysis. Extent of coronary artery disease is so different between the two groups making comparison difficult, as authors have mentioned.

Operative data also need to be presented since this study is basically looking at procedure-oriented outcome, including number of grafting, use of blood products operation time, and the completeness of revascularization.

Please clearly define ‘repeat revascularization’ in NHI database. Higher rate of repeat revascularization in off-pump group is highly suspected to be related to differences in extent of coronary artery lesion, and clear definition of ’repeat revascularization’ is critical in appropriate understanding of the obtained results.

I would propose to add propensity score matching for these analyses to define the study population for each group and compare the long-term results.

On-pump beating technique might be an important alternative procedure which provides more frequent chances of complete revascularization than off-pump CABG. My question here is whether more recent isolated CABG cases, registered after 2011, include more on-pump beating CABG cases. If so, this group also had better be highlighted in this study.

Author Response

This retrospective study highlighted the long-term clinical results of the patients under 45 years of age undergoing isolated CABG, especially comparing the outcome between those with on-pump (probably with cardiac arrest) manner (n=741) and those with off-pump manner (n=344).

Point 1 : Preoperative comorbidities regarding dyslipidemia or familial hypercholesterolemia are missing along with too many factors including history of previous PCI, NHYA/CCS class, preoperative left ventricular function, and presence of heart valve lesions and some other factors as they have mentioned in the limitation section. These missing factors should have been also considered in multivariate analysis. Extent of coronary artery disease is so different between the two groups making comparison difficult, as authors have mentioned.

Response 1

Thank you very much for the suggestion, we added the data of preoperative comorbidities such as hyperlipidemia, familial hypercholesterolemia, and prior percutaneous coronary intervention incidence in the revised table 1. In our national database we cannot obtain left ventricular ejection fraction, which is indeed the limitation of our database. We selected the patients focus on isolated coronary artery bypass disease and exclude those who have valvular heart disease in the database. By this selection, we supposed to eliminate the patient with valve lesion. However reviewer mentioned, there might be little probability that the patient who had functional mitral regurgitation and at the operation time, the mitral regurgitation was not treated surgically and was gathering in the study group. We thought that might be in a very limited number in this patient group.

Point 2: Operative data also need to be presented since this study is basically looking at procedure-oriented outcome, including number of grafting, use of blood products operation time, and the completeness of revascularization.

Response 2

Thank you for your suggestion, we added operative outcome in our 30-days outcome in the revised table 2. The operative outcome including transfusion amount, incidence of new onset Af, hospital stay and ICU stay. 

The number of grafting was added on the revised table 1.

Point 3: Please clearly define ‘repeat revascularization’ in NHI database. Higher rate of repeat revascularization in off-pump group is highly suspected to be related to differences in extent of coronary artery lesion, and clear definition of ’repeat revascularization’ is critical in appropriate understanding of the obtained results.

Response 3

About the completeness of revascularization, we used graft number/disease vessels as a marker, if the number is more than 1; we deem this is complete revascularization in our database. By doing so, the completeness of revascularization in one vessel disease was 100% in both groups. The completeness of revascularization in 2-vessels disease was 93% in off-pump group and 95% in on-pump group. The completeness of revascularization in 3-vessels disease was 91% in off-pump group and 92% in on-pump group.

We also added one paragraph in the discussion to explain this portion.

Repeat revascularization was defined by further admission intervention of coronary artery or repeat coronary artery bypass surgery 6 months after first operation. The reason we defined repeat revasculization by 6 months after first operation is to eliminate the hybrid procedures in some selective patients, because most of the hybrid procedures were performed within 3 months.

Point 4 I would propose to add propensity score matching for these analyses to define the study population for each group and compare the long-term results.

Response 4

We did propensity score matching used every variables available in Table 1 including age, gender, DM treated with insulin, hypertension, stroke, history of myocardial infarction, peripheral vascular disease, chronic obstructive pulmonary disease, implanted pacemaker, atrial fibrillation, renal replacement therapy, chronic kidney disease, normal renal function, extent of coronary artery disease (one-vessel, two-vessel and three-vessel). Besides those variables, we also use operation year as variables in PS matching. We used 0.2 as caliber of width, and used 1:1 greedy matching method, after that estimated Hazard ratio by cox model. 

PS matching model C(Concordance)- statistics=0.782

(figure can be seen in the attached file)

*The role of the c-statistic in variable selection for propensity score models. Pharmacoepidemiol Drug Saf. 2011 Mar; 20(3): 317–320. In reality C value >7 is acceptable in the real practice, our study the C(Concordance)- statistics=0.782.

After the propensity score matching, we got 232 patients in both groups. The rest of the results were the same with the original dataset in the revised manuscript. The 30-days, one-year, 5-years and 10-years survival rate was the same.

Point 5 On-pump beating technique might be an important alternative procedure which provides more frequent chances of complete revascularization than off-pump CABG. My question here is whether more recent isolated CABG cases, registered after 2011, include more on-pump beating CABG cases. If so, this group also had better be highlighted in this study.

Response 5

Thank you for the important question and suggestion. 

It is true that on-pump beating is an important alternative procedure. In our designed study, we focus only on off-pump and on-pump cardiac arrest by cardiologic solution. The on-pump beating procedures were not included in this study design. In the study period, the on-pump beating procedures account for less than 4% in all our coronary artery bypass patients, and to purified comparison about off-pump versus on-pump group, we did not use the data of on-pump beating procedures. The on-pump and off-pump coronary artery bypass trend in our registry database from 2004 to 2018 was about 30% by off-pump, and the on-pump beating procedures was around 4-8% in this period. That is a good idea, we might looking at the specific on-pump beating group in the future study.

Reviewer 2 Report

The authors conducted a retrospective analysis assessing Taiwan NHI program and extracting patients between 18 and 45 years of age undergone CABG operations in the 2001-2011 period.  The aim of the study is long-term outcome comparison between on-pump and off-pump techniques. They were able to analyze 741 young patients undergone on-pump surgery to compare them to 344 off-pump patients. The paper is well written and analysis is sound although some limitation should be highlighted:

1.    Preoperative characteristics listed in table 1 show a low risk profile for these patients. There is a higher rate of single vessel disease and lower rate of three vessels disease in the off-pump group, as expected. Apparently no other difference exists between groups. However a preoperative risk score such as EuroScore or STS score should be given to verify. No information about urgency/emergency is given and preoperative ejection fraction. Did any of these patients receive prior PCI?

2.    Outcomes at 30 days were not different between groups although important outcomes such as atrial fibrillation, blood products transfusion, ICU and Hospital length of stay were not listed. 

3.    In the long-term there is no difference in survival but off-pump patients required more frequently repeated revascularization. The authors did not elucidate if this was due to incomplete revascularization at the time of surgery or by-pass failure. As stated in the Discussion section, lack of data regarding completeness of revascularization is a major limitation for this study.  Not knowing this important data does not allow assuming a clear position. Young patients have really low CPB-related risk and therefore all should be directed towards better long-term outcome. This paper does not help understanding what should the technique of choice be.

In conclusion, the goal of this study is reasonable but the data presented is not sufficient to draw meaningful conclusion. Lack of information on preoperative patients characteristics, completeness of revascularization and reason for repeated revascularization are truly important limitation.

Author Response

Point 1.    Preoperative characteristics listed in table 1 show a low risk profile for these patients. There is a higher rate of single vessel disease and lower rate of three vessels disease in the off-pump group, as expected. Apparently no other difference exists between groups. However a preoperative risk score such as EuroScore or STS score should be given to verify. No information about urgency/emergency is given and preoperative ejection fraction. Did any of these patients receive prior PCI?

Response 1 

Thank you for the review and excellent suggestions.

Indeed, those young patients are low risk patients. Not only young in the age but also have less comorbidities compared to average CABG patient (65 years old). In our national database, we cannot obtain data regarding the EuroScore or STS score, that is the limitation of our national wide database. We added emergency/elective operation and previous PCI variable in the revised table 1.

The emergent operation in both group were similar 13.95% in off-pump group and 13.76% in on pump group (p=0.41).

The patient underwent prior percutaneous coronary intervention was 28.19% in off pump group and 26.85% in on-pump group respectively (p=0.63) 

Point 2.    Outcomes at 30 days were not different between groups although important outcomes such as atrial fibrillation, blood products transfusion, ICU and Hospital length of stay were not listed. 

Response 2

Thank you for the suggestion, we add those data in the revised table 2A, describing the operative outcome including transfusion amount, post-operative Af incidence, ICU stay and hospital stay.

We added a paragraph in the revised manuscript to describe the operative outcome. “……..The operative outcome (Table 2A) including transfusion amount, post-operative atrial fibrillation, ICU stay and hospital stay were no difference. The incidence of post-operative atrial fibrillation was 10.46% in off-pump and 13.76% in on-pump (p=0.535).”

Point 3.    In the long-term there is no difference in survival but off-pump patients required more frequently repeated revascularization. The authors did not elucidate if this was due to incomplete revascularization at the time of surgery or by-pass failure. As stated in the Discussion section, lack of data regarding completeness of revascularization is a major limitation for this study.  Not knowing this important data does not allow assuming a clear position. Young patients have really low CPB-related risk and therefore all should be directed towards better long-term outcome. This paper does not help understanding what should the technique of choice be.

Response 3

Repeat revascularization was defined by further admission intervention of coronary artery or repeat coronary artery bypass surgery 6 months after first operation. The reason we defined repeat revasculization by 6 months after first operation is to eliminate the hybrid procedures in some selective patients, because most of the hybrid procedures were performed within 3 months.

About the completeness of revascularization, we used graft number/disease vessels as a marker, if the number is more than 1; we deem this is complete revascularization in our database. By doing so, the completeness of revascularization in one vessel disease was 100% in both groups. The completeness of revascularization in 2-vessels disease was 93% in off-pump group and 95% in on-pump group. The completeness of revascularization in 3-vessels disease was 91% in off-pump group and 92% in on-pump group.

In conclusion, the goal of this study is reasonable but the data presented is not sufficient to draw meaningful conclusion. Lack of information on preoperative patients characteristics, completeness of revascularization and reason for repeated revascularization are truly important limitation.

We added more variables in pre-operative characteristics. The definition of repeat revascularization was defined and also explained the completeness of revascularization in the revised manuscript. We hoped by doing these, the conclusion and the study would be sound.

Round  2

Reviewer 2 Report

Thank you for considering my criticism, the paper is improved